# Acute Stress Effects over Time on the Gene Expression and Neurotransmitter Patterns in the Carp (*Cyprinus carpio*) Brain

**DOI:** 10.3390/ani14233413

**Published:** 2024-11-26

**Authors:** Constanze Pietsch, Paulina Pawlak, Jonathan Konrad

**Affiliations:** School of Agricultural, Forest and Food Sciences (HAFL), Bern University of Applied Sciences (BFH), 3052 Zollikofen, Switzerland; paulina.pawlak@bfh.ch (P.P.); jonathankonrad@gmx.ch (J.K.)

**Keywords:** research purposes, stressors, acute stress, handling

## Abstract

This study examined how acute stress and feed reward impact gene expression and neurotransmitter levels in different brain parts of juvenile carp. Fish were subjected to various stressors or a feed reward, and the gene expression patterns in different brain parts were measured 30, 60, and 90 min after treatments. Results showed that acute stress and feed reward significantly alter the expression of appetite-related genes, as well as neurotransmitter systems like serotonin and dopamine. These changes were observed in different brain parts, suggesting that both negative and positive stressors impact brain function. These findings contribute to a better understanding of how fish process stress and reward, which is important for improving fish welfare in aquaculture. The study highlights species-specific responses, indicating that the brain’s reaction to stress may differ between fish species.

## 1. Introduction

An increasing number of fish are used in aquaculture, and common carp especially is a very popular species in aquaculture due to its robustness [1,2]. Fish in aquaculture are frequently exposed to different stressors that can impact their well-being. Acute stress, such as that caused by handling practices, frequently occurs in fish farms. Upon a stressful event, the brain processes the information and initiates physiological processes aiding to cope with the challenge. At the brain level, the hypothalamic–pituitary–interrenal (HPI) axis is known to be involved in early stress responses. Furthermore, stress influences appetite regulation in the brain [3,4]. Two central factors that appear to exert the anorectic effects of stress in fish are the corticotropin releasing factor (CRF) and urotensin I [5]. Proopiomelanocortin, and cocaine- and amphetamine-regulated transcript (CART), have also been identified as strong anorexigenic factors in the fish brain [6,7]. Furthermore, intraperitoneal cortisol application has resulted in a dose-dependent upregulation of neuropeptide Y (*npy*) mRNA expression in the forebrain of goldfish, *Carassius auratus* [8]. Moreover, the expression of ghrelin (*ghrel*), another orexigenic factor, along with *npy*, increased in the zebrafish brain after acute stress exposure [9].

The serotonergic, dopaminergic, and opioid pathways are also involved in the stress response of fish [10,11]. For stress responses, not only serotonin (5-hydroxytryptamine, 5-ht) and dopamine, but also their metabolites 5-hydroxyindoleacetic acid (HIAA) and 3,4-dihydroxyphenylacetic acid (Dopac), are crucial [10]. Gesto et al. [12] noted that serotonergic activity precedes the increase in plasma catecholamines in rainbow trout (*Oncorhynchus mykiss*) after exposure to an acute stressor, suggesting a regulatory role of 5-ht in stress responses. 5-ht has shown anorexic effects in rainbow trout that at least partially involve CRF and urotensin I [5]. However, not only handling stress, but also social stress, appear to involve 5-ht responses in fish [13,14].

Dopamine, a neurotransmitter involved in learning and motivational behaviours, has been shown to be involved in stress responses [13]. The enzyme tyrosine hydroxylase (TH) is rate limiting in the dopamine synthesis [15], whereas dopamine receptors (DOPAR 2 and 3) realize the down-stream effects of dopamine but can also limit dopamine release [16]. Different stressors have been shown to have stimulatory effects on the dopaminergic system in teleosts [10,17,18]. Furthermore, besides its involvement in pain relief [19,20] regulating the immune system [21] or controlling feed intake [22], the opioid system is also suggested to play a role in stress responses in fish [21,23].

In the central nervous system, the neurotransmitter gamma-aminobutyric acid (GABA) mediates inhibitory effects. Furthermore, the GABA_A_ receptor activity is associated with anxiety in fish [24,25]. In addition, Li et al. [26] showed that marker genes for glutamatergic and GABAergic neurons had the highest expression levels across all brain parts in largemouth bass (*Micropterus salmoides*), thus emphasizing the importance of these neurotransmitters. Glutamate can activate the metabotropic glutamate receptor 4, an umami receptor that further interacts with NPY in Mandarin fish (*Siniperca chuatsi*), thus affecting feed intake [27]. Furthermore, Gesto et al. [28] demonstrated the influence of the arginine vasotocin pathway on stress responses in rainbow trout. This involvement has also been confirmed in other fish species [4,29,30]. Additionally, isotocin and vasotocin signaling was further suggested to be involved in the regulation of metabolic processes in the liver, influencing energy availability following air exposure [31].

Previous studies revealed that carp react immediately to different acute stressors [4]. To further elucidate the effects on brain gene regulation and neurotransmitter levels, the present study was conducted which used different sampling timepoints after each treatment. Hence, the aim of this study was to investigate the dynamics of the gene expression regulation as responses to stress exposure in four brain parts of juvenile carp.

## 2. Materials and Methods

### 2.1. Rearing Conditions

A 2000-L rearing tank was used for initial fish rearing as part of a recirculating aquarium system equipped with a bead filter and a UV lamp (FTN AquaArt AG, Rafz, Switzerland). The juvenile carp (*Cyprinus carpio*) with a mean initial total length of 13.92 cm were fed twice daily; once with thawed mosquito larvae (Ruto frozen fish food, Netherlands from Zoo Rocco, Lyss, Switzerland) in the morning between 8:00 and 8:30 a.m., as well as with commercial dry feeds in the afternoon (Aller Primo Float 3 mm, AllerAqua, Germany). From these fish, three individuals for each treatment group in duplicate were transferred to individual 90 L rearing tanks (FTN AquaArt AG, Rafz, Switzerland) connected to a recirculating aquaria system equipped with a moving bed biofilter and a UV system (FTN AquaArt AG, Rafz, Switzerland). The values from the monitoring of water temperature, oxygen levels, pH, and conductivity in the inflow have been reported in our previous study [32]. The fish were reared in this system for 3 days with continuing the feeding regime as described above. For the sampling, the control fish were taken directly from the 90-L tanks before the feeding. For the remaining treatments, the fish were either fed in the morning (feed), left unfed but received the filtered water in which the mosquito larvae were thawed (feedctr), were chased with a net for 1 min (chas), or confined with a net for 1 min (confine), or were exposed to air by netting for 1 min (air). Euthanasia was started 30, 60, or 90 min after treatment. The euthanasia was performed with an overdose of 2-phenoxyethanol (0.4 mL/L, Sigma-Aldrich, Buchs, Switzerland). After excision, the brains were stored in RNAlater^®^ (Sigma-Aldrich, Buchs, Switzerland) for at least 24 h, after which they were cut into four parts (telencephalon, hypothalamus, optic tectum, and rhombencephalon comprising the corpus cerebelli and the medulla oblongata) using a microscope (VWR^®^ VisiScope^®^ STB150, Stereomikroskop, VWR International, Dietikon, Switzerland). All experiments and procedures were approved by the local Animal Care Committee (licence no. BE69/2020) and were in accordance with the guidelines of the Swiss Council on Animal Care.

### 2.2. PCR Conditions

The primer pairs used for the gene expression studies have been published previously [4,33,34]. The primer pairs had been validated and the respective PCR products confirmed by Sanger sequencing, as described before [30]. Total RNA was extracted from the telencephalon, hypothalamus, optic tectum, and rhombencephalon separately, using RNeasy Micro Kits with on-column DNAse I treatment (Qiagen AG, Hombrechtikon, Switzerland). The RNA contents were confirmed on a plate reader (Tecan Infinite M200 Pro, Tecan Instruments, Crailsheim, Germany). Two ng of total RNA were reverse-transcribed using the High-Capacity cDNA Reverse Transcription Kit (Applied Biosystems, purchased from Thermo Fisher Scientific, Basel, Switzerland) according to the manufacturer’s protocol. The cDNA concentrations were adjusted to 50 ng/uL with nuclease-free water (Ambion^®^, purchased from Thermo Fisher Scientific, Basel, Switzerland). The PCRs were performed on a LC480 Light Cycler II (Roche, Basel, Switzerland) in 384-well plates. PCR conditions were selected as follows: 95 °C for 10 min and [95 °C, 15 s; 60 °C, 30 s] × 40 cycles. For the reference gene calculation, an open-source tool was used (RefFinder, https://www.ciidirsinaloa.com.mx/RefFinder-master/, accessed on 1 October 2024). For the telencephalon, the reference gene evaluation revealed that *ef* and *eiF4a* were the most stable genes, whereas for the remaining brain parts *citrsyn* was the optimal reference gene. After that, the gene expression values were first calculated relative to the expression of the selected reference gene, and were then further calculated as fold-changes relative to the control group. The log2-transformed relative normalized expression value of each gene was used for data modelling, as described below. The results shown in the Figure 1, Figure 2, Figure 3 and Figure 4 comprise only significant gene expression differences.

### 2.3. Neurotransmitter Measurements

Tissue homogenates from each of the four brain parts were also used to analyse neurotransmitter levels. For this, the brain parts were homogenized manually using Micro-Pistills (Omnilab-Laborzentrum GmbH & Co. KG, Bremen, Germany) and adjustment of the volume to 200 ul with nuclease-free water (Ambion^®^, purchased from Thermo Fisher Scientific, Basel, Switzerland). The commercial ELISA kits were used according to the manufacturers protocols. Serotonin, 5-ht (Cat. No. ab285243) and dopamine (Cat. No. ab285238) were analysed with ELISAs purchased from abcam.com (Cambridge, UK, website accessed on 12 July 2023). The ELISA kits for the serotonin metabolite 5-hydroxyindoleacetic acid, 5-HIAA, and the dopamine metabolite 3,4-dihydroxyphenylacetic acid, Dopac, were obtained from the CliniSciences Group (Kloten, Switzerland). For each test, 50 ul of tissue homogenate was used and incubated as described in the ELISA protocols. Since sufficient tissue was available from the optic tectum, also gamma-aminobutyric acid, gaba (Cat. No. KA4912, purchased from Lucerna-Chem, Luzern, Switzerland) and glutamate (Cat. No. KA1909, purchased from Lucerna-Chem, Luzern, Switzerland) were measured in this brain part. The neurotransmitter levels were calculated as ng neurotransmitter per mg brain tissue.

### 2.4. Calculations and Statistics

The gene expression and neurotransmitter data were evaluated based on mixed models with a fully Bayesian approach using the *brms* package [35] in R studio (Version 1.2.1335) for each stressor separately. For this, gene-specific random effects for the constants, gene-specific random effects for the group differences, as well as animal-specific random effects for the constants were considered. The model fit was assessed visually by using QQ plots for the distribution of y and y_rep_. Improved handling of possible outliers was achieved by using posterior predictive checks based on the Markov Chain Monte Carlo (MCMC) approximation, whereby the simulated replicated data (y_rep_) under the fitted model were compared with the observed data (y), as has been explained previously by Burren and Pietsch [36]. For each model, the point estimators, their standard errors of the mean (SEM), the credibility intervals, and the posterior predictive *p* values were calculated. The significance values were based on Wald χ2-statistics for generalized linear models and F-statistics for mixed models, and the estimated marginal means were calculated including the corresponding 95% credible intervals. A *p* value of <0.05 was considered statistically significant. Furthermore, the log2-transformed gene expression datasets together with the neurotransmitter values were subjected to a principal component analysis (PCA) to show data patterns that typically correspond to the distinct treatments. The PCA was performed in R studio, as described earlier [33]. More details on the contribution of each of the two main components to the total variance are given for the individual treatment groups in the Results section.

## 3. Results

### 3.1. Gene Expression Patterns

In the telencephalon (Figure 1), the expression of genes, such as *npy* and *cart*, was up-regulated in all treatments, whereas the expression of *cck* was down-regulated. In addition, *ghrel* was not significantly changed by chasing. However, dopamine receptors were not influenced in the confined fish, although the remaining treatment groups showed changes in expression of at least one of the two dopamine receptors that were investigated. Moreover, *gabaa* was not involved in the stress responses in the telencephalon of chased and air-exposed fish. The feed reward and the feed control treatment were characterized by the absence of an *iso pre* response compared with the remaining treatments. Furthermore, *il-1β* was only increased in air-exposed fish 60 min after the treatment, whereas the remaining fish showed decreased or no changes of the *il-1β* expression after treatment. In contrast, confined fish showed no changes of the *5ht-r* expression. A detailed description of the different expression values in the telencephalon can be found in Appendix A.

For the hypothalamus of carp, the expression patterns of *cart*, *grp*, *il-1β*, *serotr*, and the vasotocin receptors appeared to be quite similar for all treatment groups (Figure 2). In contrast, the feed reward and feed control fish showed a prolonged activation *nmu*, *npy*, and *opio d*. The expression of *opiod* was also decreased only in chased fish, whereas chased and air-exposed fish showed a decreased expression of *mtor*, and only fish belonging to the feed control and the confinement group showed up-regulation of *iso pre*. In addition, only air-exposed fish showed a decreased *gabaa* expression 60 min after treatment. A detailed description of the different expression values in the hypothalamus can be found in Appendix A.

In the optic tectum of carp, the expression patterns of many genes appeared to be quite similar for confined and air-exposed fish. Obvious similarities of the gene expression patterns have also been observed for the feed-rewarded and the feed control fish, which was less obvious for the chased animals. The feed control fish were the only treatment group that showed increased expression of *dopar 2*, whereas only air-exposed animals revealed decreased *cck* and increased *ghrel* expression 60 min after treatment. In contrast to the remaining treatment groups, confined fish showed no changes of the *npy* expression. A detailed description of the different expression values in the optic tectum can be found in Appendix A.

In the rhombencephalon of carp, the expression of fewer changes in the gene expressions were observed for feed-rewarded fish compared with the remaining treatments. The air-exposed fish showed some similarity with the gene expression patterns of confined fish, but fish sampled 60 min showed contrasting expression levels.In addition, the dopamine receptors and the isotocin-related genes showed no changes in the fish belonging to the feed reward, feed control, and the chasing groups. Moreover, *mtor* was only reduced in air-exposed fish. A detailed description of the different expression values in the rhombencephalon can be found in Appendix A.

### 3.2. Neurotransmitter Levels

In the telencephalon of feed-rewarded fish, the 5ht levels were significantly higher 30 min after treatment compared with the remaining groups (*p* = 0.004; Table 1). In addition, feed-rewarded fish showed higher Dopac levels 30 min after treatment compared with the controls and fish 90 min after treatment (*p* ≤ 0.016). No significant differences in the neurotransmitter levels between the feed control fish were observed. Chased fish showed higher 5ht levels 90 min after treatment than in the controls (*p* = 0.044) and higher Dopa levels 60 min after treatment (*p* = 0.008). In addition, chased fish showed a significantly lower Dopac level in the telencephalon 60 min after treatment compared with fish sampled after 30 min (*p* = 0.024). The Dopa levels in confined fish were higher 60 min and 90 min after treatment than in the other two treatment groups (*p* ≤ 0.042). Finally, the Dopa levels in the telencephalon of air-exposed fish was higher 60 min after treatment compared with the controls (*p* = 0.004).

In the hypothalamus, the 5ht levels were significantly different between feed-rewarded fish sampled 90 min after treatment and the controls (*p* < 0.001, Table 2). No significant differences in the neurotransmitter levels of the hypothalamus between the feed control fish or chased fish compared with the control animals were observed. In confined fish, the 5ht levels 60 min and 90 min after treatment were found to be lower than in the controls or in fish sampled after 30 min (*p* < 0.038). In addition, confined fish showed higher HIAA levels 90 min after treatment compared with the controls or fish sampled after 60 min (*p* ≤ 0.020). Furthermore, the 5ht levels in the air-exposed fish were lower and the levels of HIAA higher 30 min and 90 min after treatment compared with the controls (*p* ≤ 0.038).

In the optic tectum of feed-rewarded, feed control and confined fish no differences were observed, except for a higher glutamate level in the fish 90 min after feed control treatment compared with fish sampled after 60 min (*p* = 0.040, Table 3). In the optic tectum of chased fish, the 5ht levels were lower 90 min after treatment compared with the remaining treatments (*p* ≤ 0.020).

Finally, the optic tectum of air-exposed fish showed lower 5ht levels 90 min after treatment compared with the fish sampled after 60 min (*p* = 0.028), and the 60 min and 90 min revealed higher HIAA levels than in the other two groups (*p* ≤ 0.026).

In the rhombencephalon of feed-rewarded fish, all treated fish showed higher 5ht levels than the controls (*p* ≤ 0.014, Table 4), and lower Dopa levels 30 min after treatment compared with the controls and higher levels 90 min after treatment compared with the fish sampled 30 min and 60 min after treatment (*p* ≤ 0.020). In the rhombencephalon of feed control fish, the 5ht levels were higher 60 min and 90 min after treatment than in the controls (*p* ≤ 0.002). In addition, these fish showed lower Dopa levels 60 min and 90 min after treatment compared with fish sampled after 30 min (*p* ≤ 0.014). Chased fish showed higher 5ht levels in all treated fish compared with the controls (*p* ≤ 0.034), and lower Dopa levels 60 min and 90 min after treatment than in fish sampled after 30 min (*p* ≤ 0.032). The same brain part showed higher 5ht levels 60 min and 60 min after confinement compared with the other two groups (*p* ≤ 0.024), and higher HIAA levels 60 min after treatment compared with the remaining treatment groups (*p* ≤ 0.012). Finally, the 5ht levels in this brain part were higher in fish sampled 60 min after air exposure than in the controls and fish sampled after 90 min (*p* ≤ 0.012).

### 3.3. Principal Component Analyses

The results from the principal component analyses (PCA) for all genes and brain parts are available in Appendix A. Based on these results, the two most important genes for each brain regulation pathway were selected and used for additional PCA. The PCA for the telencephalon explained on average 72% of the variation in the data with the first two components (Appendix A). Combined with the neurotransmitter levels, the PCA explained on 64.5% to 88.4% of the variation in the data of the differently treated fish with the first two components (Figure 5). For this final set of genes in the telencephalon, *ox* and *vasor 1* and *2* were identified as genes with a strong relevance for the expression patterns in this brain part in feed rewarded fish (Figure 5), whereas feed control fish had *vasor 2, ox* and *npy* as the most contributing genes. Chased fish also had *vasor 2* as the most contributing gene in the telencephalon, but also *th* and *il-1β*. Confined fish showed high influence of *th, vasor2* and *ox*, whereas air-exposed fish *ghrel, gastr* and *it-r1* as the most contributing genes in the telencephalon.

The PCA for the hypothalamus explained on average 67.7% of the variation in the data with the first two components (Appendix A). Combined with the neurotransmitter levels, the PCA explained on 57.1% to 72.4% of the variation in the data of the differently treated fish with the first two components (Figure 6). Furthermore, a high contribution of *ghrel, ox* and *it-r1* to the outcome of gene expressions in the final set of genes in the hypothalamus of feed-rewarded fish was noted (Figure 6), but *ghrel*, *vasor 2* and *serotr* were among the most contributing genes in this brain part of feed control fish. Moreover, the genes with the highest influence in the hypothalamus of chased and confined fish were an appetite-related gene, *ghrel*, but also *it-r1* and *il-1β* (Figure 6). Air-exposed fish showed high influence of *ghrel* and *it-r1* in the hypothalamus, but also of *gastr*.

The PCA for the optic tectum explained on average 86.74% of the variation in the data with the first two components (Appendix A). Combined with the neurotransmitter levels, the PCA explained on 70.59% to 78.93% of the variation in the data of the differently treated fish with the first two components (Figure 7). In the optic tectum of feed-rewarded carp, the appetite-related gene *cart* as well as *it-r1* and *dopar 3* had a strong ability to explain the variability in the data sets after application of the different stressors (Figure 7). Feed control fish showed high contribution of *prolr*, *vasor 2* and *ghrel*, whereas chased fish showed high influence of *it-r1*, but also of the neurotransmitters GABA and glutamate. The genes with the highest influence in the optic tectum of confined fish were the appetite-related gene, *grp*, but also *vasor 2* and *dopar 3* (Figure 7). The *dopar 3* was also important in the optic tectum of air-exposed with together with *ghrel* and *it-r1*.

Finally, the PCA for the rhombencephalon explained on average 84.4% of the variation in the data with the first two components (Appendix A). Combined with the neurotransmitter levels, the PCA explained on 77.4% to 85.4% of the variation in the data of the differently treated fish with the first two components (Figure 8). The rhombencephalon showed the strongest influence of the genes *vasor 2*, *npy* and *it-r1* in feed-rewarded fish (Figure 8), whereas *it-r1* together with the Dopac and HIAA levels were also important in feed control fish. Moreover, *it-r1* and Dopac in parallel to the expression of *dopar 2* were also important in chased fish. In contrast, *it-r1*, *prolr* and *ghrel* showed high contribution in the rhombencephalon of confined fish. The genes with the highest influence in the rhombencephalon of air-exposed fish were the appetite-related genes, *ghrel* and *nmu*, but also *dopar3* (Figure 8).

## 4. Discussion

In this study, acute negative stressors and feed reward were used to examine the different effects of the stressors on the juvenile carp brain. Our previous study on zebrafish [30] demonstrated that feed reward as a positive stressor and air exposure as a negative stressor affect gene expression in the different gene regions differently. In the current study, similar stressors that typical occur during to regular work with fish were tested, over different time points. The results indicate that carp react to the same stressors differently than zebrafish, but the difference in the gene expression patterns of fish that received the feed reward or were treated by air exposure or confinement was obvious in several brain parts of carp. The results, furthermore, showed that in some brain parts, e.g., the rhombencephalon of feed-rewarded fish, the response to the stressor is visible for 30 to 60 min, whereas other stressors resulted in a prolonged stress response for at least 90 min. These findings support the assumption of a species-specific plasticity of neuronal activities upon stress which is also known to define certain coping styles in fish [37].

### 4.1. Appetite-Related Genes

Distress is known to affect the food intake of fish through the interaction with NPY and CRF neurons [38,39]. However, central administration of NPY increased the food intake in goldfish [40]. In the present study, the expression of *npy* was increased at different time points in the telencephalon after the different negative stressors, whereas 60 and 90 min after receiving a feed reward, the expression of *npy* in telencephalon and hypothalamus was higher than in the control group. Interestingly, a similar effect had already been observed after 30 min after the feed control treatment. Furthermore, an effect of increased glutamate levels in the feed control group did not influence the *npy* expression in the present study. It may be hypothesized that an effect of glutamate on the *npy* expression as described for Mandarin fish [27] would have needed more time in carp, since the glutamate increase in the optic tectum was observed 90 min after treatment and the current investigations did not use samplings after that timepoint to evaluate further effects on the brain gene expression. In zebrafish, an increase of the *npy* mRNA levels in the hypothalamus were observed, not only after application of negative stressors, but also 90 min after receiving feed [32]. In addition, the expression of *ghrel* in zebrafish was increased in the optic tectum at each time point after confinement and air exposure, as well as 30 min after feeding [32]. Similarly, Cortés et al. [9] described an increase of *npy* and *ghrel* expression in the zebrafish brain after exposure to acute stress. It was assumed that an increase in expression of orexigenic genes following a stressful challenge is necessary to counteract the anorexigenic effects of stress [9].

In zebrafish, the expression of *cart* showed lower levels after air exposure in the optic tectum and the rhombencephalon [32]. In the present study, *cart* expression was decreased in optic tectum 30 min and 90 min after air exposure. In the rhombencephalon, this effect was observed only after 60 min, and a decreased expression was noted 30 and 90 min after the stressor. Similar effects were caused by confinement. CART is involved in several physiological functions including energy homeostasis, appetite regulation, and responses to stress [7,41]. However, *cart* in zebrafish was downregulated in the optic tectum 30 min, and in the telencephalon 90 min also after receiving a feed reward [32]. This effect, however, was not observed in this study. It has been suggested that CART may have two-directional effects on appetite, influenced by satiety and starvation [42].

### 4.2. Genes of the Serotonergic Pathway

Expecting a feed reward that is received later than expected or not at all leads to higher aggression of Atlantic salmon (*Salmo salar*) towards conspecifics [43]. Furthermore, aggression, especially in social contexts, but also stress responses can be influenced by actions of serotonin and dopamine pathways [13,44]. Interestingly, chasing increased *5ht-r* expression and decrease *5ht receptor b* expression in zebrafish 60 min after treatment in the hypothalamus [28] suggesting a downregulation of this circuit. In zebrafish, also downregulation of the *crf-r2* and *mr* were observed in the hypothalamus 90 min after chasing [28]. The role of the serotonin pathway in stress responses appears to be very complex, affecting numerous body functions [11,45,46]. The telencephalon is rich in serotonergic neurons [47] and, due to the presence of amygdala-like regions, it is furthermore assumed to play a role in complex processing of sensory inputs in fish [48,49,50,51]. The quick serotonin system response to acute stress may also suggests its role as a mediator between stress perception and the initial catecholamines and corticoid release [12].

### 4.3. Involvement of Dopamine in Stress Responses

Both stimulatory and inhibitory effects of stress on the dopaminergic system have been shown in fish [10,11,17]. In a study conducted on rainbow trout (*Oncorhynchus mykiss*), acute stress did not affect the dopamine system in the hypothalamus, however, its increased activity was observed in the telencephalon [12]. Furthermore, Gesto et al. [12] confirms that the magnitude of the dopaminergic system response is stressor- and brain region-dependent. This is also supported by the current study as significant activation of the dopamine system was observed in the telencephalon after air exposure and chasing. Feed rewarding increased the expression of the *dopar 2* in the telencephalon after 60 and 90 min, followed by an increased expression of the *th* gene. Interestingly, increased expression of *dopar2* was also noted in telencephalon and hypothalamus after feed control, for which an appetitive reward was omitted, supporting the involvement of the dopamine system in the anticipatory and reward-linked responses [52,53].

### 4.4. Measurement of Other Pathways and Neurotransmitters

A study conducted on gilthead sea bream (*Sparus aurata*) suggests that telencephalon the primary brain part involved in the hedonic regulation of food intake in fish [22]. However, increased opioid activity was observed in the current study not only in the telencephalon, but also in the hypothalamus and optic tectum, indicating that not only the telencephalon, but also other brain parts may contribute to hedonic feed intake regulation. Furthermore, the opioid system activity varied with different treatments across the brain regions, supporting its role in modulating stress effects in fish [21], as well as suggesting that the responses are specific to different stressors.

The GABA system is known to be involved in the inhibition of neuronal activity. While the GABA_A_ receptor is associated with anxiety in fish [24,25], its role in increasing feeding activity has been shown in other vertebrates [54,55]. For instance, in mice, feed intake increases GABA levels in the lateral hypothalamus [56]. Furthermore, in goldfish a crosstalk between the GABA_A_ receptor and orexin has been observed [57], indicating that GABA may impact feeding behavior also in fish. In the present study, an increase in the *gabaa* expression was observed in the telencephalon, and optic tectum after feed rewarding, further suggesting its role in feeding in fish. Interestingly, *gabaa* expression decreased in the hypothalamus and rhombencephalon 60 min after air exposure and increased 30 and 90 min in the telencephalon, optic tectum and rhombencephalon after confinement and/or chasing indicating stress-specific regulatory mechanisms. Given the importance of GABA and glutamate in the fish brain [26], these two neurotransmitters should gain more attention in future experiments. Although the neurotransmitter glutamate has been shown to interact with NPY in Mandarin fish [27] this interaction was not observed in the current study on carp. Further studies should be conducted to evaluate the exact timing of the interaction of glutamate and appetite genes in carp.

The expression of *it-r1* in all four brain parts was frequently among the most contributing genes in the PCA, what follows findings of other studies, suggesting the involvement of the isotocin pathways in stress responses in fish [29,31]. Furthermore, the PCA showed that the inclusion of the neurotransmitter levels in the analyses did not improve the cumulative variance that could be explained in the telencephalon, the hypothalamus and the optic tectum. However, the analyses for the rhombencephalon were improved by the integration of the neurotransmitter levels into the PCA.

## 5. Conclusions

This study examined the effects of short-term exposure to negative stressors and feed reward on juvenile carp brains, revealing stress-specific responses. Key findings include the increased *npy* expression in response to stressors and feed rewards, and significant alterations in serotonergic and dopaminergic pathways. Additionally, the study highlights the roles of the opioid and GABA systems in stress and feeding regulation. PCA analysis emphasized the importance of neurotransmitter levels in the rhombencephalon. In summary, the current investigations show that even short-term stress affects the appetite regulations as well as further pathways in the carp brain for at least 1.5 h. These insights contribute to a better understanding of stress and reward mechanisms in fish, with potential applications in aquaculture welfare practices. They, for example, should result in reduced handling of fish in aquaculture and definition of strict feeding protocols for establishing a routine for the fish since this may avoid prolonged activation of stress responses in farmed fish.

## Figures and Tables

**Figure 1 animals-14-03413-f001:**
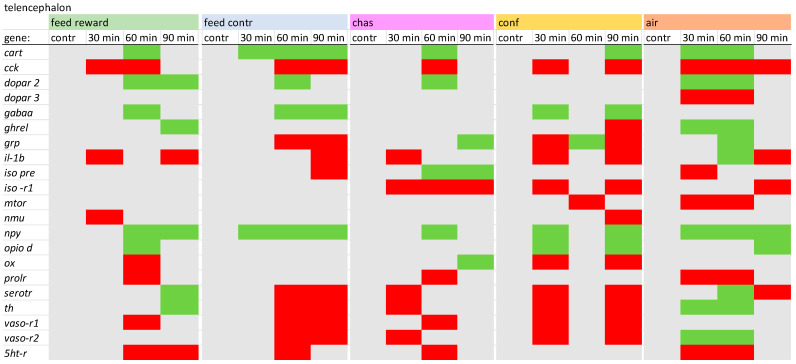
Overview of the gene expression profiles in the telencephalon of control fish (contr) and fish after the treatment (30 min; 60 min; and 90 min) after the different treatments as described in the Materials and Methods section: feed rewarding, feed control, chasing, confinement and air exposure, *n* = 6 per treatment, red = down-regulated, green = up-regulated, grey = not changed, *p* > 0.05).

**Figure 2 animals-14-03413-f002:**
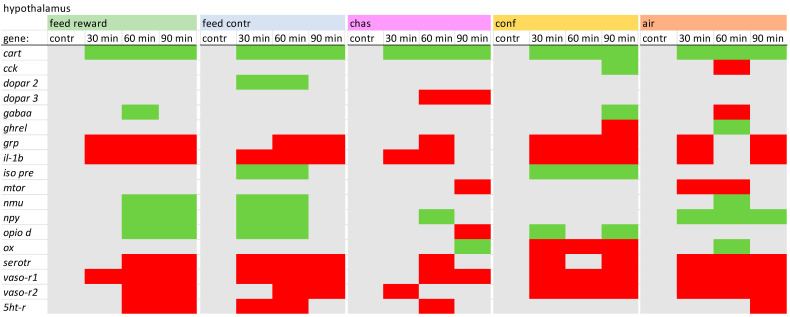
Overview of the gene expression profiles in the telencephalon of control fish (contr) and fish after the treatment (30 min; 60 min; and 90 min) after the different treatments as described in the Materials and Methods section: feed rewarding, feed control, chasing, confinement and air exposure, *n* = 6 per treatment, red = down-regulated, green = up-regulated, grey = not changed, *p* > 0.05).

**Figure 3 animals-14-03413-f003:**
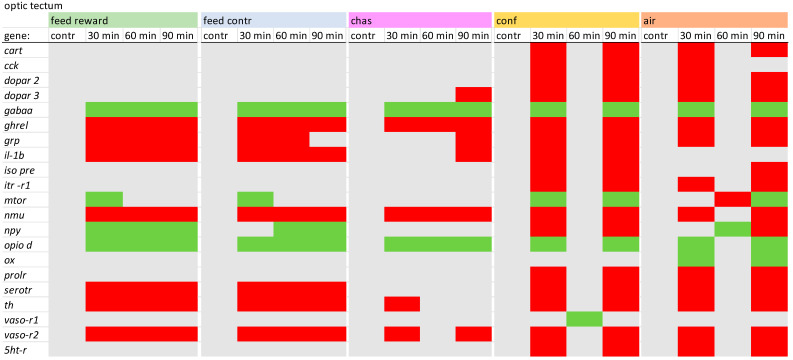
Overview of the gene expression profiles in the optic tectum of control fish (contr) and fish after the treatment (30 min; 60 min; and 90 min) after the different treatments as described in the Materials and Methods section: feed rewarding, feed control, chasing, confinement and air exposure, *n* = 6 per treatment, red = down-regulated, green = up-regulated, grey = not changed, *p* > 0.05).

**Figure 4 animals-14-03413-f004:**
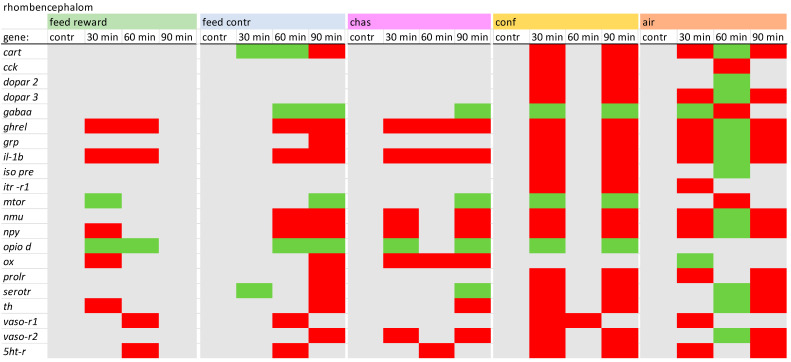
Overview of the gene expression profiles in the rhombencephalon of control fish (contr) and fish after the treatment (30 min; 60 min; and 90 min) after the different treatments as described in the Materials and Methods section: feed rewarding, feed control, chasing, confinement and air exposure, *n* = 6 per treatment, red = down-regulated, green = up-regulated, grey = not changed, *p* > 0.05).

**Figure 5 animals-14-03413-f005:**
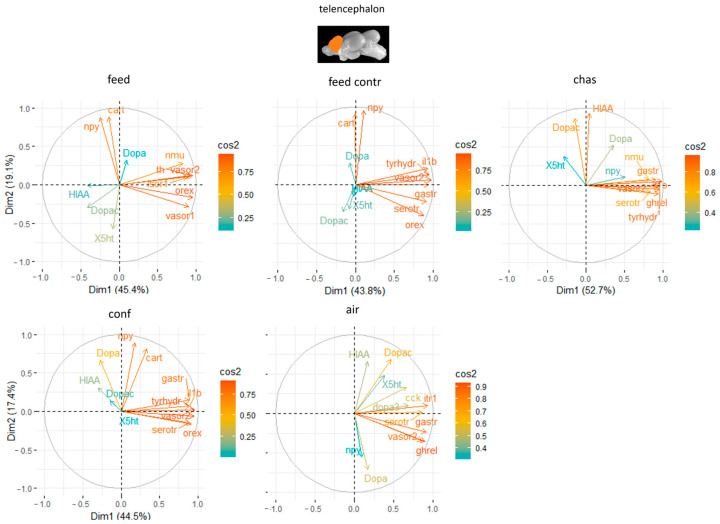
Gene expression analysis with principal component analyses (PCA) for the most contributing genes in each of the telencephalon showing their representation on the factor map as cos2 values, whereby the numbers next to Dim1 and Dim2 indicate the percentage of the variance in the datasets that is explained by the first two components of the PCA of fish 0, 30 min, 60 min and 90 min after treatment; *n* = 6 per treatment.

**Figure 6 animals-14-03413-f006:**
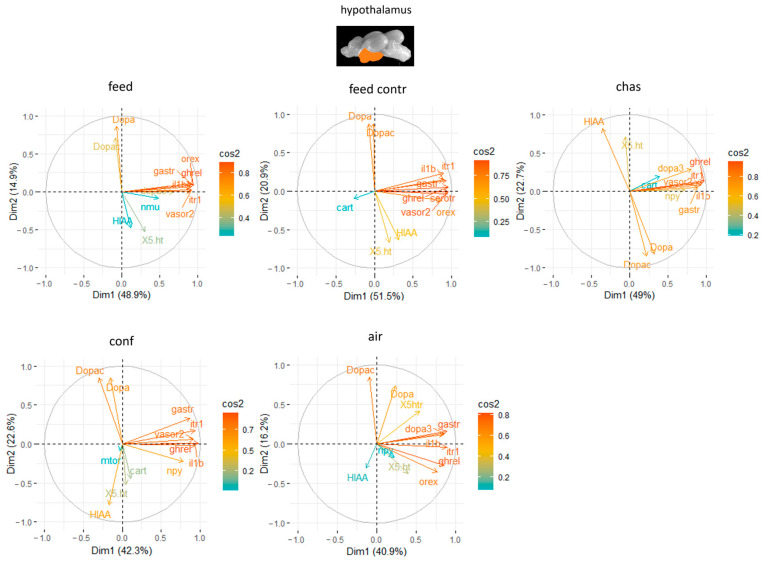
Gene expression analysis with Principal Component Analyses (PCA) for the most contributing genes in each of the hypothalamus showing their representation on the factor map as cos2 values, whereby the numbers next to Dim1 and Dim2 indicate the percentage of the variance in the datasets that is explained by the first two components of the PCA of fish 0, 30 min, 60 min and 90 min after treatment; *n* = 6 per treatment.

**Figure 7 animals-14-03413-f007:**
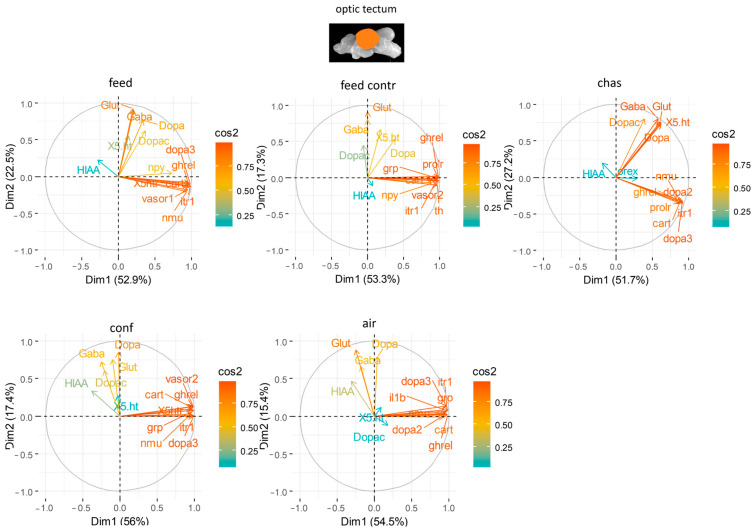
Gene expression analysis with principal component analyses (PCA) for the most contributing genes in each of the optic tectum showing their representation on the factor map as cos2 values, whereby the numbers next to Dim1 and Dim2 indicate the percentage of the variance in the data sets that is explained by the first two components of the PCA of fish 0, 30 min, 60 min, and 90 min after treatment; *n* = 6 per treatment.

**Figure 8 animals-14-03413-f008:**
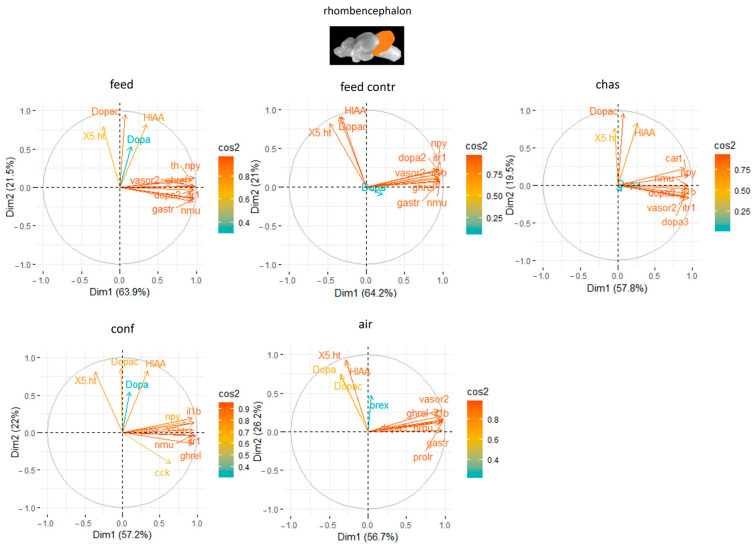
Gene expression analysis with principal component analyses (PCA) for the most contributing genes in each of the rhombencephalon showing their representation on the factor map as cos2 values, whereby the numbers next to Dim1 and Dim2 indicate the percentage of the variance in the datasets that is explained by the first two components of the PCA of fish 0, 30 min, 60 min, and 90 min after treatment; *n* = 6 per treatment.

**Table 1 animals-14-03413-t001:** Neurotransmitter levels in the telencephalon; *n* = 6 per treatment, means of groups with the same letters are not significantly different from each other, *p* > 0.05.

	5-ht [ng]		HIAA [ng]		Dopa [ng]		Dopac [ng]	
	per mg Tissue		per mg Tissue		per mg Tissue		per mg Tissue	
	Mean	sd	Mean	sd	Mean	sd	Mean	sd
contr	2.92 ^a^	2.26	1.13	0.66	4.50	1.22	2.89 ^a^	1.40
feed 30	10.38 ^b^	7.44	1.22	0.33	7.58	3.46	3.49 ^b^	0.74
feed 60	1.60 ^a^	1.50	1.21	0.55	9.09	5.77	2.63 ^a,b^	1.01
feed 90	1.86 ^a^	1.54	1.04	0.24	16.99	16.88	2.16 ^a^	0.33
feed contr 30	7.60	10.79	1.60	0.60	7.90	2.81	2.96	1.75
feed contr 60	1.23	0.54	1.17	0.54	8.42	2.34	2.95	0.17
feed contr 90	1.59	1.88	0.93	0.11	6.22	1.68	2.87	0.48
chasing 30	32.90 ^a,b^	21.53	1.30	0.38	5.32 ^a,b^	1.59	3.44 ^a,b^	0.83
chasing 60	5.59 ^a,b^	4.02	1.23	0.59	7.91 ^b^	2.54	2.51 ^a,c^	1.27
chasing 90	6.32 ^b^	11.30	1.17	0.23	6.11 ^a,b^	3.05	2.75	0.47
confine 30	16.41	12.11	1.12	0.31	5.92	1.79 ^a^	2.88	0.38
confine 60	20.90	23.91	1.45	0.34	6.84	1.57 ^b^	3.04	0.99
confine 90	5.83	5.01	1.30	0.44	7.58	1.60 ^b^	2.92	0.57
air 30	4.07	4.89	1.14	0.32	16.93	12.70	2.20 ^a,b^	0.92
air 60	13.81	15.81	1.48	0.23	9.05	2.47	3.62 ^b^	1.08
air 90	3.44	4.29	0.76	0.30	12.33	7.23	1.99 ^a,b^	1.16

**Table 2 animals-14-03413-t002:** Neurotransmitter levels in the hypothalamus; *n* = 6 per treatment, means of groups with the same letters are not significantly different from each other, *p* > 0.05.

	5-ht [ng]		HIAA [ng]		Dopa [ng]		Dopac [ng]	
	per mg Tissue		per mg Tissue		per mg Tissue		per mg Tissue	
	Mean	sd	Mean	sd	Mean	sd	Mean	sd
contr	166.80 ^a^	191.27	1.66	0.96	9.71	14.03	1.24	0.90
feed 30	257.84 ^a,b^	126.46	2.20	0.32	0.91	0.69	0.43	0.13
feed 60	9.34 ^a,b^	8.18	1.88	1.08	3.97	8.10	0.56	0.31
feed 90	18.32 ^b^	13.72	3.40	1.76	0.65	0.32	0.84	0.54
feed contr 30	116.72	181.41	1.64	1.19	27.84	28.82	2.00	1.31
feed contr 60	80.22	155.33	2.82	1.62	5.11	5.86	0.77	0.29
feed contr 90	108.05	142.42	2.51	1.04	0.66	0.84	0.55	0.32
chasing 30	92.23	189.54	1.91	1.26	9.75	14.69	1.31	0.60
chasing 60	55.15	108.93	1.71	0.76	13.69	20.58	1.35	1.45
chasing 90	149.42	204.49	2.28	0.95	5.61	12.09	0.92	0.50
confine 30	169.16 ^a^	182.06	2.30 ^a,b^	0.68	1.17	1.56	0.42	0.09
confine 60	101.02 ^b^	149.34	1.33 ^a^	0.59	18.87	18.09	1.19	0.88
confine 90	18.63 ^c^	11.80	2.64 ^c^	1.12	1.99	1.40	0.64	0.21
air 30	34.71 ^b^	11.98	3.04 ^b^	0.74	0.75	0.70	0.69	0.36
air 60	110.78 ^a^	131.34	2.37 ^a,b^	1.86	4.13	5.37	0.70	0.28
air 90	29.98 ^b^	14.38	3.29 ^b^	1.35	1.60	1.88	0.84	0.36

**Table 3 animals-14-03413-t003:** Neurotransmitter levels in the optic tectum; *n* = 6 per treatment, means of groups with the same letters are not significantly different from each other, *p* > 0.05.

	5-ht [ng]		HIAA [ng]		Dopa [ng]		Dopac [ng]		gaba [ng]		glutamate	
	per mg Tissue		per mg Tissue		per mg Tissue		per mg Tissue		per mg Tissue		per mg Tissue	
	Mean	sd	Mean	sd	Mean	sd	Mean	sd	Mean	sd	Mean	sd
contr	16.16 ^a^	1.54	0.98 ^a^	0.24	0.52	0.13	0.01	0.01	1902.48	360.05	11.16 ^a,b^	1.50
feed 30	14.90	1.97	14.13	14.81	0.48	0.10	0.01	0.01	1873.11	201.68	10.75	0.97
feed 60	11.66	6.63	1.27	0.38	0.50	0.15	0.01	0.01	1834.39	346.84	10.21	4.23
feed 90	16.32	3.68	21.83	26.22	0.56	0.30	0.03	0.03	2454.21	1118.14	15.32	8.14
feed contr 30	11.18	8.31	1.16	0.17	0.53	0.17	0.01	0.01	2000.98	305.32	11.23 ^a,b^	1.65
feed contr 60	10.37	5.73	6.39	5.73	0.51	0.06	0.01	0.01	1756.94	219.50	9.12 ^a^	1.48
feed contr 90	11.10	2.81	1.69	0.85	0.45	0.12	0.01	0.01	1955.07	279.80	10.87 ^b^	1.48
chasing 30	18.50 ^a^	4.93	0.96	0.31	0.40	0.35	0.02	0.02	1936.18	271.30	11.45	1.89
chasing 60	26.05 ^a^	23.72	14.92	13.47	0.87	0.83	0.03	0.05	2654.27	2246.78	16.23	18.14
chasing 90	14.01 ^b^	5.23	1.45	0.85	0.47	0.17	0.01	0.01	1825.36	390.47	10.14	2.45
confine 30	19.31	3.80	15.25	15.94	0.52	0.16	0.03	0.06	1918.08	418.26	11.07	1.20
confine 60	13.04	4.41	0.98	0.44	0.54	0.20	0.02	0.01	1890.17	307.61	11.29	3.11
confine 90	11.75	5.27	1.75	0.81	0.42	0.22	0.02	0.01	1838.34	168.26	9.75	2.79
air 30	17.07 ^a^	3.91	0.98 ^a^	0.28	0.47	0.26	0.06	0.09	1737.92	209.60	10.75	1.95
air 60	17.09 ^a,b^	13.69	1.51 ^b^	0.20	0.52	0.07	0.16	0.11	1786.71	71.52	10.66	0.63
air 90	14.23 ^a,c^	3.18	2.12 ^b^	1.04	0.44	0.14	0.14	0.13	1905.94	190.67	11.07	1.20

**Table 4 animals-14-03413-t004:** Neurotransmitter levels in the rhombencephalon; *n* = 6 per treatment, means of groups with the same letters are not significantly different from each other, *p* > 0.05.

	5-ht [ng]		HIAA [ng]		Dopa [ng]		Dopac [ng]	
	per mg Tissue		per mg Tissue		per mg Tissue		per mg Tissue	
	Mean	sd	Mean	sd	Mean	sd	Mean	sd
contr	5.01 ^a^	5.77	1.63 ^a^	0.57	0.69 ^a^	0.38	0.37	0.33
feed 30	9.12 ^b^	5.49	1.51	0.64	0.30 ^b^	0.38	0.33	0.21
feed 60	20.50 ^c^	8.53	1.61	0.79	0.74 ^a,b^	0.97	0.43	0.27
feed 90	12.71 ^c^	7.98	1.59	0.26	1.77 ^c^	1.47	0.43	0.15
feed contr 30	15.60 ^a,b^	10.87	1.78	0.52	1.09 ^a,b^	0.54	0.40	0.21
feed contr 60	24.38 ^b^	20.69	2.09	20.69	0.55 ^a,c^	0.42	0.61	0.45
feed contr 90	10.90 ^c^	5.80	1.35	0.25	0.61 ^a,c^	0.70	0.33	0.08
chasing 30	6.69 ^b^	2.86	1.41	0.18	0.90 ^a,b^	0.40	0.22	0.12
chasing 60	22.08 ^c^	1.70	1.48	0.17	0.67 ^a,c^	0.55	0.42	0.06
chasing 90	16.82 ^b^	7.01	1.77	0.32	0.64 ^a,c^	0.53	0.52	0.25
confine 30	6.66 ^a^	2.40	1.35 ^a^	0.32	0.75	0.47	0.26	0.10
confine 60	19.24 ^b^	13.88	1.98 ^b^	0.31	2.68	4.79	0.46	0.25
confine 90	20.00 ^b^	0.70	1.41 ^a^	0.24	1.06	0.51	0.30	0.09
air 30	14.12 ^a,b^	10.16	1.65	0.70	1.09	0.50	0.73	0.74
air 60	24.27 ^b^	26.99	2.23	1.47	2.88	4.74	0.59	0.58
air 90	4.78 ^a^	2.22	1.41	0.20	1.45	0.39	0.26	0.11

## Data Availability

The original data presented in the study are available in this repository: https://figshare.com/articles/dataset/PCR_Carp_V2_Acute_Stress/27342339?file=50086011, accessed 5 November 2024.

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
