# Peer review of "Acute Stress Effects over Time on the Gene Expression and Neurotransmitter Patterns in the Carp (Cyprinus carpio) Brain"

_animals, 2024, doi:10.3390/ani14233413_

Round 1
Reviewer 1 Report
Comments and Suggestions for Authors
A brief summary:
This MS by Pietsch et al. carried out a stress trial with juvenile common carp to investigate transcriptomic differences in four brain parts in response to acute negative stressors and feed reward, focusing on appetite-related genes, serotonergic and dopaminergic pathways, and other involved systems, at 30, 60, and 90 min after treatments. The treatments showed pronounced effects on the gene expression patterns across brain parts compared to control fish. Notably, npy expression increased in the telencephalon following negative stressors and feed reward, suggesting a stress-coping mechanism by promoting food intake. Cart expression in carp showed varying responses, indicating species-specific regulation of appetite and stress. Serotonergic and dopaminergic pathways were also affected, with alterations in the respective receptors’ expression, confirming their roles in stress and reward processing. Principal component analyses revealed that neurotransmitter levels in the different brain parts contribute to the explained variance.
Overall, I thought this was a well-executed study in a system with limited previous knowledge of this level on this specific topic. I appreciated their multi-faceted approach (ELISA and Molecular Biology) and the time course involved in this work and think it add significant merit to their work. I do think that their overall conclusions were related to results. I think this is good work, but should undergo minor revision before acceptance.
1) Specific comments
1. Introduction
In “Introduction”, the author introduced mainly focused on effects on the gene expression and neurotransmitter factors such as “dopamine” “serotonin” and “gamma-aminobutyric acid”. However, there is little information on carp (Cyprinus carpio). Why the author chose this species of fish to do this study? What about the aquaculture output and advantage of carp in Switzerland? The author should show information that it is necessary that they choose this fish for this study. What is the meaning and help of this study for the carp aquaculture industry in Switzerland? I think this information is very important to show the readers in background, especially the readers from other countries culturing the carp.
Furthermore, I find that the authors had published a similar paper using zebrafish as target species in Frontiers in Physiology, and they also cited it in the references 26: Pietsch, C., Konrad, J., Wernicke von Siebenthal, E., and Pawlak, P. (2024). Multiple faces of stress in the zebrafish (Danio rerio) brain. Front Physiol 15. doi: 10.3389/fphys.2024.1373234
In the references 35, Pietsch, C., Pawlak, P., and Konrad, J. (2024). Acute stress effects overtime on the stress axis in the carp (Cyprinus carpio) brain. Aquac Res (submitted).
I cannot find this paper because it was not published.
I worried if there is some repeation and similarity between these papers. Because the methods are similar and only the species change from “zebrafish” to “carp”. And also, the detected gene and index may be different between these studies. I hope, if possible, check the repetition rate between these papers.
Finally, I think the author's research ideas, methods, results, and discussion are very skillful and professional in this area. If the repetition rate meets the requirements for publication, I suggest minor revision before acceptance.
Comments on the Quality of English LanguageThe English could be improved to more clearly express the research.
Author Response
Notably, npy expression increased in the telencephalon following negative stressors and feed reward, suggesting a stress-coping mechanism by promoting food intake. Cart expression in carp showed varying responses, indicating species-specific regulation of appetite and stress. Serotonergic and dopaminergic pathways were also affected, with alterations in the respective receptors’ expression, confirming their roles in stress and reward processing. Principal component analyses revealed that neurotransmitter levels in the different brain parts contribute to the explained variance.
Thank you for sending us your comments on the manuscript. All changes to the manuscript have been highlighted in yellow.
Comment 1: In “Introduction”, the author introduced mainly focused on effects on the gene expression and neurotransmitter factors such as “dopamine” “serotonin” and “gamma-aminobutyric acid”. However, there is little information on carp (Cyprinus carpio). Why the author chose this species of fish to do this study? What about the aquaculture output and advantage of carp in Switzerland? The author should show information that it is necessary that they choose this fish for this study. What is the meaning and help of this study for the carp aquaculture industry in Switzerland? I think this information is very important to show the readers in background, especially the readers from other countries culturing the carp.
Response 1: Thank you very much for this comment. The relevance of carp species in aquaculture has been added to the introduction.
Comment 2: Furthermore, I find that the authors had published a similar paper using zebrafish as target species in Frontiers in Physiology, and they also cited it in the references 26: Pietsch, C., Konrad, J., Wernicke von Siebenthal, E., and Pawlak, P. (2024). Multiple faces of stress in the zebrafish (Danio rerio) brain. Front Physiol 15. doi: 10.3389/fphys.2024.1373234
In the references 35, Pietsch, C., Pawlak, P., and Konrad, J. (2024). Acute stress effects overtime on the stress axis in the carp (Cyprinus carpio) brain. Aquac Res (submitted). I cannot find this paper because it was not published.
I worried if there is some repeation and similarity between these papers. Because the methods are similar and only the species change from “zebrafish” to “carp”. And also, the detected gene and index may be different between these studies. I hope, if possible, check the repetition rate between these papers.
Response 2: Thank you, the similarity between the articles has been checked by the journal Animals and we have removed phrases that may have been too similar. In addition, it may be important to note that zebrafish – although being a cyprinid species as well – shows differences in gene expression patterns after stress and is not used in aquaculture, but in research and as a pet species. Taking this into account, it makes sense to analyse similar things in the two species. And the results support our approach since we received different results from the different studies. In summary, this confirms that stress responses are highly species-specific and the more trials are conducted the more we learn about the species-specific characteristics. The comparison to different species has been made a bit bigger in the manuscript to emphasize this aspect.
Finally, I think the author's research ideas, methods, results, and discussion are very skillful and professional in this area. If the repetition rate meets the requirements for publication, I suggest minor revision before acceptance.
Reviewer 2 Report
Comments and Suggestions for Authors
The article " Acute stress effects over-time on the gene expression and neu rotransmitter patterns in the carp (Cyprinus carpio) brain" is of great interest to readers. Overall, this work has elicited the following comments:
1. In the Introduction section, the authors thoroughly and interestingly substantiate the relevance of this work. This section of the work deserves high marks, it is scientifically substantiated and supported by relevant references to the literature.
2. The Materials and Methods section needs significant revision. It is necessary to provide the working PCR protocol, according to which the authors performed the corresponding work. References to previously published works are not enough. It is necessary to describe in detail and step by step the procedure of the enzyme immunoassay, which the authors measured the levels of neurotransmitters. It is also necessary to indicate how the authors carried out the calibration and the brands of the equipment that was used in the quantitative determination of neurotransmitters.
3. It is necessary to clarify the criteria for conducting statistical analysis. This section of materials and methods also needs to be detailed and presented in detail.
4. The Results section contains a lot of descriptive information. The authors always present the quantitative data they obtained without analyzing them. In the presented version of the article, the Results section is also unsatisfactory. It is necessary to analyze and summarize the obtained material. It is best to present the digital material in the form of tables, highlighting the important values. The authors should find a more presentable form of presenting the research results, and highlighting the most important results, focusing on them. At present, this section of the article is difficult to perceive and analyze.
5. The Discussion section is interesting, however, the authors discuss the voluminous material they obtained too narrowly. The data established by the authors can be discussed more broadly in the context of issues of neuronal plasticity, the regulatory effect of neurotransmitters on the processes of eating behavior, and finally, a comparison of broad neurohistological data obtained on different fish species to discuss the dynamics of neurotransmitter systems under the conditions of behavioral experiments.
Comments on the Quality of English LanguageEnglish needs editing
Author Response
The article " Acute stress effects over-time on the gene expression and neurotransmitter patterns in the carp (Cyprinus carpio) brain" is of great interest to readers. Overall, this work has elicited the following comments:
Thank you for sending us your comments on the manuscript. All changes to the manuscript have been highlighted in yellow.
- In the Introduction section, the authors thoroughly and interestingly substantiate the relevance of this work. This section of the work deserves high marks, it is scientifically substantiated and supported by relevant references to the literature.
Response 1: Thank you, we have also added some information about carp as an important species in aquaculture to this section to emphasize the relevance of the investigations.
- The Materials and Methods section needs significant revision. It is necessary to provide the working PCR protocol, according to which the authors performed the corresponding work. References to previously published works are not enough. It is necessary to describe in detail and step by step the procedure of the enzyme immunoassay, which the authors measured the levels of neurotransmitters. It is also necessary to indicate how the authors carried out the calibration and the brands of the equipment that was used in the quantitative determination of neurotransmitters.
Response 2: Thank you. The details for the lab work have been added.
- It is necessary to clarify the criteria for conducting statistical analysis. This section of materials and methods also needs to be detailed and presented in detail.
Response 3: The analyses have now been described in detail. This also includes the statistical part and the modelling.
- The Results section contains a lot of descriptive information. The authors always present the quantitative data they obtained without analyzing them. In the presented version of the article, the Results section is also unsatisfactory. It is necessary to analyze and summarize the obtained material. It is best to present the digital material in the form of tables, highlighting the important values. The authors should find a more presentable form of presenting the research results, and highlighting the most important results, focusing on them. At present, this section of the article is difficult to perceive and analyze.
Response 4: Thank you. We totally agree. Therefore, the descriptive part was moved to the supplement and the results section was designed new, showing only up-regulation and downregulation of genes without mentioning too many details. This allowed to concentration further on the differences between the treatment groups.
- The Discussion section is interesting, however, the authors discuss the voluminous material they obtained too narrowly. The data established by the authors can be discussed more broadly in the context of issues of neuronal plasticity, the regulatory effect of neurotransmitters on the processes of eating behavior, and finally, a comparison of broad neurohistological data obtained on different fish species to discuss the dynamics of neurotransmitter systems under the conditions of behavioral experiments.
Response 5: Thank you. Details explaining this have been added now.
Comments on the Quality of English Language: English needs editing
Response 6: Thank you. The language has been checked and improved, where necessary.
Reviewer 3 Report
Comments and Suggestions for Authors
The manuscript, "Acute stress effects over-time on the gene expression and neurotransmitter patterns in the carp (Cyprinus carpio) brain," investigates the impact of acute stress and feed rewards on gene expression and neurotransmitter levels in various brain regions of juvenile carp. This study provides insights into the physiological mechanisms of stress response and reward processing in fish, particularly focusing on species-specific reactions that could influence aquaculture practices for welfare improvement. The strength of the manuscript lies in the following points:
- The study combines gene expression analysis and neurotransmitter measurements across multiple brain regions, offering a holistic view of the stress response.
- Understanding stress and reward in fish can help optimize handling practices in aquaculture, reducing stress-related morbidity and improving fish welfare.
- The study utilizes well-defined experimental protocols with appropriate statistical analyses, including mixed models and PCA, enhancing the robustness of the findings.
Areas for Improvement
The authors mention species-specific responses but could expand on the implications of these findings in comparison to other species, like zebrafish. Discussing interspecies differences in stress and reward pathways would provide context and improve the study's impact.
Some sections on methodology, especially sample handling and brain dissection, could benefit from further clarification for reproducibility.
While serotonin and dopamine's roles are discussed, more information on other neurotransmitters, such as GABA and glutamate, would provide a fuller picture of how multiple pathways contribute to stress responses.
Briefly explain the choice of Bayesian mixed models and how they contribute to interpreting gene expression differences.
The study would benefit from a section dedicated to discussing how these findings might influence specific aquaculture practices, such as changes in handling or feeding protocols.
The paper could benefit from references to other fish species’ stress studies to underscore the relevance and uniqueness of the findings in carp.
The manuscript is well-structured and provides significant insights into the neurophysiological impacts of stress and reward on juvenile carp. By improving clarity in methodology, expanding on comparative discussions, and highlighting practical applications, this study could be a valuable contribution to the field of aquatic animal welfare and stress physiology.
Author Response
Thank you for sending us your comments on the manuscript. All changes to the manuscript have been highlighted in yellow.
Areas for Improvement:
The authors mention species-specific responses but could expand on the implications of these findings in comparison to other species, like zebrafish. Discussing interspecies differences in stress and reward pathways would provide context and improve the study's impact.
Response 1: Thank you. This aspect has now gained more attention.
Some sections on methodology, especially sample handling and brain dissection, could benefit from further clarification for reproducibility.
Response 2: Thank you. More details and descriptions have been added.
While serotonin and dopamine's roles are discussed, more information on other neurotransmitters, such as GABA and glutamate, would provide a fuller picture of how multiple pathways contribute to stress responses.
Response 3: Sorry, that this has not been included in the previous version of the manuscript. Especially, the glutamate pathway has now gained much more attention.
Briefly explain the choice of Bayesian mixed models and how they contribute to interpreting gene expression differences.
Response 4: The Bayesian models are in our opinion a good choice for analyzing so many genes in parallel. Other statistical methods may not be able to integrate so many features and factors at the same time without losing information or losing statistical power. In addition, inferential statistics allow us to calculate the probability of a hypothesis – in our case the difference of a certain gene expression after the fish received a stressful treatment. And probabilities and predictions are increasingly important to evaluate the relevance of biological observations.
The study would benefit from a section dedicated to discussing how these findings might influence specific aquaculture practices, such as changes in handling or feeding protocols.
Response 5: Thank you. These aspects have been added to the conclusion.
The paper could benefit from references to other fish species’ stress studies to underscore the relevance and uniqueness of the findings in carp.
Response 6: A number of publications on other fish species has been added.
The manuscript is well-structured and provides significant insights into the neurophysiological impacts of stress and reward on juvenile carp. By improving clarity in methodology, expanding on comparative discussions, and highlighting practical applications, this study could be a valuable contribution to the field of aquatic animal welfare and stress physiology.
Response 7: Thank you. We have hopefully improved the manuscript sufficiently.
Round 2
Reviewer 2 Report
Comments and Suggestions for Authors
The revised version of the manuscript has been significantly improved compared to the original version. The authors in the response letter and accordingly in the revised version of the manuscript have satisfied almost all the requirements that were proposed to improve the quality of the manuscript. The manuscript can be recommended for publication in the journal.
Comments on the Quality of English Language.